# Multi-Trait Genomic Prediction of Yield-Related Traits in US Soft Wheat under Variable Water Regimes

**DOI:** 10.3390/genes11111270

**Published:** 2020-10-28

**Authors:** Jia Guo, Jahangir Khan, Sumit Pradhan, Dipendra Shahi, Naeem Khan, Muhsin Avci, Jordan Mcbreen, Stephen Harrison, Gina Brown-Guedira, Joseph Paul Murphy, Jerry Johnson, Mohamed Mergoum, Richanrd Esten Mason, Amir M. H. Ibrahim, Russel Sutton, Carl Griffey, Md Ali Babar

**Affiliations:** 1Department of Agronomy, University of Florida, Gainesville, FL 32611, USA; neojiaguo@gmail.com (J.G.); jkazrc@UFL.EDU (J.K.); sumitp@ufl.edu (S.P.); dshahi@ufl.edu (D.S.); naeemkhan@ufl.edu (N.K.); mavci@ufl.edu (M.A.); jcmcbreen@ufl.edu (J.M.); 2School of Plant Environment and Soil Sciences, Louisiana State University, Baton Rouge, LA 70803, USA; sharrison@agctr.lsu.edu; 3USDA-ARS, North Carolina State University, Raleigh, NC 27607, USA; Gina.Brown-Guedira@ars.usda.gov; 4Department of Crop and Soil Sciences, North Carolina State University, Raleigh, NC 27607, USA; paul_murphy@ncsu.edu; 5Department of Crop and Soil Sciences, University of Georgia, Griffin, GA 32223, USA; jjohnso@uga.edu (J.J.); mmergoum@uga.edu (M.M.); 6Department of Crop Soil and Environmental Sciences, University of Arkansas, Fayetteville, AR 72701, USA; esten@uark.edu; 7Department of Soil and Crop Sciences, Texas A&M University, College Station, TX 77843, USA; aibrahim@tamu.edu (A.M.H.I.); r-sutton@tamu.edu (R.S.); 8School of Plant and Environmental Sciences, Virginia Tech, Blacksburg, VA 24061, USA; cgriffey@exchange.vt.edu

**Keywords:** genomic prediction, multi-trait model, multi-environment genomic best linear unbiased predictor, Bayesian multi-trait multi-environment model, Bayesian multi-output regressor stacking model, deep learning multi-trait multi-environment model

## Abstract

The performance of genomic prediction (GP) on genetically correlated traits can be improved through an interdependence multi-trait model under a multi-environment context. In this study, a panel of 237 soft facultative wheat (*Triticum aestivum* L.) lines was evaluated to compare single- and multi-trait models for predicting grain yield (GY), harvest index (HI), spike fertility (SF), and thousand grain weight (TGW). The panel was phenotyped in two locations and two years in Florida under drought and moderately drought stress conditions, while the genotyping was performed using 27,957 genotyping-by-sequencing (GBS) single nucleotide polymorphism (SNP) makers. Five predictive models including Multi-environment Genomic Best Linear Unbiased Predictor (MGBLUP), Bayesian Multi-trait Multi-environment (BMTME), Bayesian Multi-output Regressor Stacking (BMORS), Single-trait Multi-environment Deep Learning (SMDL), and Multi-trait Multi-environment Deep Learning (MMDL) were compared. Across environments, the multi-trait statistical model (BMTME) was superior to the multi-trait DL model for prediction accuracy in most scenarios, but the DL models were comparable to the statistical models for response to selection. The multi-trait model also showed 5 to 22% more genetic gain compared to the single-trait model across environment reflected by the response to selection. Overall, these results suggest that multi-trait genomic prediction can be an efficient strategy for economically important yield component related traits in soft wheat.

## 1. Introduction

From 2007 to 2050, farmers will need to increase the production of cereals by 60% to feed over 9.5 billion people in the world [1]. Meanwhile, this must be done under a continuously changing environment due to extreme weather conditions, land pressure, and increased energy use [2,3,4,5]. Hence, it is of paramount importance to renovate breeding technologies to increase food production while mitigating pressure on the environment. Genomic prediction (GP), originally proposed by Meuwissen et al. [6], is becoming widely used by plant breeders in recent years to advance breeding progress. The availability of high-throughput phenotyping and cost-effective genotyping technologies were the most important factors for the successful and effective implementation of GP in plant breeding [7,8]. With the help of improved statistical models, GP can be augmented to be more accurate and applicable in various scenarios in plant breeding such as multi-trait and multi-environment schemes.

Compared to traditional marker-assisted selection, GP does not require prior knowledge about a few, large-effect quantitative trait loci, since all genotypic markers are curated in training prediction models [6]. Essentially, the genomic estimated breeding value (GEBV) of individuals can be calculated using genome-wide molecular markers and phenotypic data. Then, a predictive model is constructed using a training set of individuals with known phenotypic and genotypic information. In the validation set of individuals, GEBV are calculated based on their genotypic information and the previously constructed model. Then, the accuracy of the predictive models is evaluated by cross-validation approaches within and among environment. Several empirical studies have shown that GP is effective in accelerating breeding cycles and improving genetic gains per unit of time in major crops [8,9,10].

A key component in GP is the choice and optimization of models that are used to estimate the marker effect. Statistical models with different capacities are required to handle the ever-growing magnitude of phenotypic and genotypic data. The number of predictor variables (*p*) is much larger than the number of observations (*n*) due to the improved availability of genotypic data compared to phenotypic. As such, penalized GP models such as ridge-regression best linear unbiased prediction (rrBLUP), least absolute shrinkage and selection operator (LASSO), and elastic net are employed to control the trade-offs between lack of fit and model complexity [6,11,12]. Bayesian methods are also used for parameterization in GP models [13,14,15]. As most of these models are univariate and focus on predicting one dependent variable at the time, a multivariate model incorporating associations among several dependent variables can improve the power of predictive models [16,17,18]. A multivariate model is also effective in predicting continuous variables closely associated with each other, which is a common situation for quantitative traits such as grain yield and nutrition content in cereal crops [16,19,20]. In large-scale plant breeding programs, multi-environmental trials add another layer of challenges in dissecting information from genotype × environment interaction (G×E). A three-way genomic model for evaluating the prediction accuracy of trait × genotype × environment could advance GP in modern plant breeding programs.

A few multivariate and/or multi-environment predictive models have been proposed for binary, ordinal, count, and continuous traits. Several studies applied a Bayesian-based model for multi-trait analysis and observed improved accuracies compared to single-trait analysis [16,19,21]. Burgueño et al. [22] and López et al. [23] extended the single-environment model to a multi-environment–multi-trait context and reported a significant improvement in GP model accuracy. In empirical field experiments, Montesinos-López et al. [24] and Guo et al. [25] observed that prediction models that incorporated hyperspectral data or other physiological traits (canopy temperature, membrane thermostability, chlorophyll content, stay green, and rate of senescence) and spectrum/trait by environment interaction terms were more accurate than those that did not. Guo et al. [25], Crain et al. [26], and Krauss et al. [27] also reported improved prediction accuracies using a multi-environment model relative to a single-environment model. Two Bayesian-based mixed multi-trait models, Bayesian Multi-trait Multi-environment (BMTME) and Bayesian Multi-output Regressor Stacking (BMORS) models were proposed by Montesinos-Lopez et al. [28,29]. The BMTME model assesses the variance–covariance structure among trait, genotype, and environment, and it jointly predicts multiple traits evaluated in multiple environments [23]. The BMORS model first calculates genomic best linear unbiased predictions (GBLUP) for each trait and then corrects accuracy in a secondary model using the prediction of the first-stage GBLUP model [23,29]. An improved version of the BMTME model was proposed by Montesinos-Lopez et al. (Montesinos-López et al. 2019b), which was equipped with optimization algorithms for efficiently applying the software to real data. Deep learning (DL) algorithms have led to success in bioinformatics research due to its versatility and flexibility [30]. One of the DL algorithms, neural networks (NNs), has showed comparable prediction accuracy to statistical models for complex human and animal traits [31,32,33]. A few studies have reported the performance of DL algorithms in plant genomic prediction. Liu and Wang [34] indicated that NNs had higher prediction accuracies compared to Bayesian or ridge regression-based methods using a set of soybean data. Ma et al. [35] compared NNs to a GBLUP model in a large set of wheat data and reported a better performance for NNs in terms of higher phenotypic value in top selected individuals and lower sensitivity to outliers. Montesinos-Lopez et al. [36] proposed a multi-trait deep learning model and compared it to the GBLUP model using maize (*Zea mays* L.) and wheat data, which showed higher prediction accuracies for NNs when the genotypic by environmental (G × E) effect was ignored, while lower prediction accuracies were observed when G×E effect was involved. In addition to model selection, the genetic structure of the trait, marker density, sample size, and composition of training population (TP) and validation population (VP) are also important factors in GP accuracy [19,37,38,39,40,41]. 

Yield component traits such as harvest index (HI), spike fertility (SF: ratio of grain number per spike to chaff weight per spike), and thousand grain weight (TGW) play important roles in the determination of grain yield (GY) in wheat. Strong genetic correlations were observed among GY, HI, SF, and TGW under different environmental conditions [42,43,44,45,46]. Thousand grain weight is usually a highly inheritable trait that positively contributes to GY [47,48]. Increasing the grain number through maximizing the partitioning of assimilates (e.g., carbohydrate) to grain instead of a non-grain part of the spike is a noble and effective approach to increase grain yield [49,50,51]. Therefore, manipulation of the grain number and spike chaff dry weight is a potential avenue for yield increase in wheat, which is supported by several field studies [46,52,53,54,55]. In addition to the significant genetic correlation with the performance, SF demonstrated strong genetic variations in advanced breeding lines and early generation breeding populations in wheat [46,56,57]. Guo et al. [25] observed high prediction accuracy (0.3–0.5) for fertile spikelet number and spike length while observing low prediction accuracy (<0.2) for SF and spikelet density (ratio of spikelet number per spike to spike length) in spring wheat. In another study using two doubled haploid wheat populations, moderate to low accuracies were observed for grain number per spike in controlled (0.10–0.42) and osmotic stress (0.27–0.46) conditions [58]. Improvement of these yield component traits is an ideal solution for enhancing sink capacity in wheat. Currently, there is no information available on multi-trait genomic prediction for GY, HI, SF, and TGW in wheat. This approach could provide an accurate prediction for jointly improving grain yield-related traits in wheat. The proposed study will provide critical information in the development of wheat germplasm through optimized yield components traits using GP. Therefore, the objectives of this study were to (1) estimate the genetic correlations among GY, HI, SF, and TGW in a multi-environment scenario, (2) compare the prediction accuracies of single- against multi-trait models under a multi-environment context, and (3) estimate the response to selection (RS) of grain yield from single- and multi-trait models under a multi-environment context. 

## 2. Materials and Methods 

### 2.1. Site Description

The experiment was conducted over two growing seasons from 2016 to 2018 at Citra and Quincy, Florida (Table 1). Citra is characterized by sandy soil with loam at 20–80 inches with low water-holding capacity, whereas Quincy has well-drained loamy soil with higher water-holding capacity than Citra. Citra had moderate precipitation (212–447 mm) during 2016–2018, moderate humidity, and temperature rise above >30 °C multiple times during the grain-filling stages. Quincy received higher precipitation (582–625 mm), had high humidity, and experienced relatively fewer episodes of high temperatures (>30 °C) during the same period. Experiments were planted in between mid-November to the first week of December.

### 2.2. Plant Genetic Material and Experimental Design

The genetic material used for the study consisted of 240 (237 + 3 checks) facultative soft wheat genotypes selected from the Gulf Atlantic Wheat Nursery (GAWN). The genotypes were developed by public wheat breeding programs (North Carolina State University, Texas A&M, Louisiana State University, University of Georgia, University of Arkansas, and Virginia Tech) targeting for the south and southeastern regions of the USA. The panel is referred to as GAWN panel. The genotypes in the panel generally require a short duration of cold treatment to satisfy the vernalization for flower induction. The panel was evaluated at two locations in Florida: Citra and Quincy during 2016–2018 (two years). To induce terminal drought stress at Citra, irrigation was stopped 2 weeks before anthesis (GS60) until maturity. Contrary to that, 1–2 supplemental irrigations were applied when needed in Quincy. Citra is considered to be a drought-stressed environment, and Quincy is moderately drought stressed. All trials were planted in six-row plots (3 m length × 1.5 m width) using a seeding rate of 100 kg h^−1^. The GAWN panel was planted in an incomplete block augmented design with repeated checks (AGS2000, PI 656845; SS8641, PI 674197; Jamestown, PI 653731) in each block with 237 unreplicated new entries [59]. Three repeated checks are widely adapted and cultivated throughout the southeastern US. To control foliar and glume diseases, fungicides were sprayed three times. Herbicides were sprayed to control weeds as required. Fertilizers were applied through irrigation for best management practices for proper growth and yield. 

### 2.3. Traits Measurement

Phenotypic data for days to heading (DTH), grain yield (GY), harvest index (HI), spike fertility (SF), and thousand grain weight (TGW) were collected. Days to heading (GS 59) were collected using the Zadoks scale [60]. GY was measured by harvesting all six rows using a small plot harvester and was calculated by dividing the total grain weight by plot area, adjusted to 12% moisture, and converted to t ha^−1^. To measure SF, ten random spikes were sampled from the field at physiological maturity, dried for 72 h at 60 °C, and threshed by a single head thresher to determine chaff weight (the non-grain part of a spike), which was calculated as the difference between total spike dry weight and spike grain weight. Spike fertility (SF) was calculated as a ratio of grains m^−2^ to spike chaff weight m^−2^ [44]. Grain m^−2^ was obtained by using seeds from SF sample (grains per spike) multiplied to the number of spikes m^−2^. To get spike number m^−2^, we harvested tillers at maturity from 0.5 m^2^ middle two rows and counted and later converted to m^2^. The HI was calculated as the ratio of grain weight m^−2^ to total dry matter m^−2^. TGW was measured by weighing 1000 grains counted through a seed counter (Seedburo Equipment Co., Chicago, IL, USA). 

### 2.4. Phenotypic Data Analysis

The best linear unbiased estimates (BLUEs) and standard errors were calculated for DTH, GY, HI, SF, and TGW using the following equation assuming genotype as a fixed effect and environment and block as random effects:*Y’_ijk_ = μ + Gg_j_+ E_i_ +B_i(k)_ + GgE_ji_ + e_ijk_*(1)
where *Y_ijk_* is the observed value; *µ* was the general mean; $Gg_{j}$ is the genotypic effect (*j* = 1 to 223); $E_{i}$ is the environment effect (*i* = 1 to 4, corresponding to Citra 2017, Citra 2018, Quincy 2017, and Quincy 2018); $B_{j\left(k\right)}$ is the block effect (*k* = 1 to 12; N[0,σB2]) nested within the *i*^th^ environment; $GgE_{ji}$ is the *j*^th^ genotype by *i*^th^ environment interaction effect; and $e_{ijk}$ is the random error (N[0,σe2]). Block and environmental effects and error are commonly modeled to follow independent normal distributions [61]. To evaluate the influence of phenology, DTH was included as an additional fixed effect in model (1) for all following analyses. The broad sense heritability (*H*^2^) from each environment was calculated using the following formula, *H*^2^ = (σ^2^_G_)/(σ^2^_G_ + σ^2^_e_), where σ^2^_G_ and σ^2^_e_ were variances due to genotype and error, respectively. Genotype and block were considered as random effects. In order to estimate variance values, we used the following model below:*Y’_ij_ = μ + Gg_j_ + B_k_ + e_jk_.*(2)

Pearson correlation analyses among four phenotypic traits were also calculated. 

### 2.5. Genotypic Data Analysis

Fresh green seedling leaf tissues of each line were used to get genomic DNA through the LGC Genomics Oktopure robotic extraction platform along with Sbeadex magnetic microplate reagent kits. Genotyping by sequencing (GBS) was performed using Illumina HiSeq 2500 after double digestion of genomic DNA with *Pstl* and *Msel* restriction enzymes [38]. SNP calling was carried out using TASSEL-GBS v5.2.49 [62,63]. The Illumina platform generated short reads were aligned using Burrows-Wheeler Aligner v0.7.17-r1188 to the Chinese Spring IWGSC RefSeq v1.0 wheat reference sequence [64]. Pre and post imputation filtering were used for retaining biallelic SNPs and removing SNP with missing data >50%, with minor allele frequencies <5%. We also remove genotypes with >85% missing data. Then, missing data were imputed using Beagle 5.1, and later, data was re-filtered to remove SNPs with minor allele frequency (MAF) <5% or heterozygous call frequency of <10%. A Fisher’s exact test was used to test if the SNP alleles at each site were independent in a population of inbred lines, as described by Poland et al. [65]. The SNPs were assumed to be allelic in the population if the null hypothesis of independence for the two alleles was rejected (*α* = 0.001). This procedure typically lowers heterozygous calls due to sequencing errors, genome duplications, and homologous sequences on different genomes [38,65,66]. In the final genomic dataset, a total of 27,957 SNPs remained.

### 2.6. Prediction Models

Three statistical models including the Multi-environment Genomic Best Linear Unbiased Predictor (MGBLUP), Bayesian Multi-trait Multi-environment (BMTME) model, and Bayesian Multi-output Regressor Stacking (BMORS), and two deep learning (DL) models including Single-trait Multi-environment Deep Learning (SMDL) and Multi-trait Multi-environment Deep Learning (MMDL) were compared for predicting GY, HI, SF, and TGW. 

#### 2.6.1. Multi-Environment Genomic Best Linear Unbiased Predictor (MGBLUP) Model

According to Montesinos-López et al. [24,67], a brief summary of three statistical models are presented in the following sections. A univariate linear mixed model is often used for accounting for effects of environment and environment × genotype interaction:(3)$$Y_{ij}=E_{i}+G_{j}+GE_{ij}+\varepsilon_{ij}$$
where ***Y****_ij_* is the best linear unbiased estimate (BLUE) of predicted trait for *j*^th^ genotype in *i*^th^ environment; $E_{i}$ is the environment effect (*i* = 1 to 4, corresponding to Citra 2017, Citra 2018, Quincy 2017, and Quincy 2018); $G_{j}$ is the genetic main effect (*j* = 1 to 223); the genetic main effect is assumed as a joint distribution of genotype effect with a multivariate normal distribution $G={\left(G_{1},\ldots ,G_{j*}\right)}^{T}\sim MN\left(0,\;\sigma_{G}^{2}G\right)$, where $\sigma_{G}^{2}$ denotes the genomic variance and **G** represents the genomic relationship matrix; the **G** matrices were calculated as $G=\frac{XX^{\prime }}{p}$, where ***X*** is a matrix of the centered and standardized SNP marker matrix and *p* is the number of SNP markers; $GE_{ji}$ is the *j^t^*^h^ genotype by *i*^th^ environment interaction effect; the term $GE_{ji}$ was assumed to have a multivariate normal distribution, that is $GE_{ji}={\left(GE_{11},\ldots ,GE_{ji}\right)}^{T}\sim MN\left(0,\;\left(Z_{g}GZ_{g}^{T}\right)\#\left(Z_{E}G_{E}^{T}\right)\sigma_{GE}^{2}\right)$ where $Z_{g}$ and $Z_{E}$ are incidence matrices for the vector of genomics and environment effects, and $\sigma_{GE}^{2}$ is the variance component for $GE_{ji}$; $\varepsilon_{ij}$ is a random residual associated with the *j*^th^ line in the *i*^th^ environment distributed as $N\left(0,{\;\sigma }^{2}\right)$ where $\sigma^{2}$ is the residual variance.

#### 2.6.2. Bayesian Multi-Trait Multi-Environment (BMTME) Model 

For the BMTME model, a matrix-variate normal distribution is assumed denoted as $M\sim NM_{n\times p}\left(H,\;\Omega ,\Sigma \right)$. The (np×1) random vector vec(M) is distributed as multivariate normal as $N_{np}\left(vec\left(H\right),\Sigma \;⊗\;\Omega \right)$; **H** is a n×p location matrix, **Σ** is a p×p first covariance matrix, and **Ω** is a n×n second covariance matrix. n is the number of genotypes, and p is the number of SNPs; vec(.) and ⊗ are the standard vector operator and Kronecker product, respectively. Therefore, a BMTME model is defined as follows:(4)$$Y=X\beta +Z_{1}b_{1}+Z_{2}b_{2}+E$$
where ***Y*** is the vector of multivariate responses of n × L, with L being the number of predicted traits and n=J×I, where J denotes the *j*^th^ genotype and I denotes the *i*^th^ environment, ***X*** is a vector of n × I, β is of order I × L; $Z_{1}$ is of order of n × J, $b_{1}$ is of order J × L and represents the genotype × trait interaction; $Z_{2}$ is a vector of order n × IJ, $b_{2}$ is a vector of order IJ ×L and represents the genotype × environment × trait interaction. Vector $b_{1}$ is assumed under a matrix-variate normal distribution as $NM_{J\times L}\left(0,\;G^{\prime },\Sigma_{t}\right)$, where **G^′^** denotes the genomic relationship matrix; the **G** matrices were calculated as $G^{\prime }=\frac{WW^{\prime }}{p}$, where W is a matrix of the centered and standardized SNP marker matrix of order J × p, and p is the number of SNP markers; and $\Sigma_{t}$ is a unstructured genetic covariance matrix of traits of order L × L, $b_{2}$ is assumed under a matrix-variate normal distribution as $NM_{JI\times L}\left(0,\Sigma_{E}⊗G^{\prime },\Sigma_{t}\right)$, where $\Sigma_{E}$ is an unstructured covariance matrix of order I × I and E is the matrix of residuals of order n × L with $E\sim NM_{n\times L}\left(0,\;I_{n},R_{e}\right)$, where $R_{e}$ is the unstructured residual covariance matrix of traits of order L × L. Genetic correlations between phenotypic traits and environments were calculated as $r_{G\left(a,b\right)}=\frac{\sigma_{G\left(a,b\right)}}{\sqrt{\sigma_{G\left(a\right)}^{2}\sigma_{G\left(b\right)}^{2}}}$, where $\sigma_{G\left(a,b\right)}$ is the covariance of traits *a* and *b*; $\sigma_{G\left(a\right)}^{2}$ is the genotypic variance of trait *a*; and $\sigma_{G\left(b\right)}^{2}$ is the genotypic variance of trait *b*.

#### 2.6.3. Bayesian Multi-Output Regressor Stacking (BMORS) Model 

The BMORS model is a two-stage predictive model originally proposed by Spyromitros-Xioufis et al. [68,69]. In the first stage, single-trait GBLUP models are established for each trait according to model (3). In the second training stage, the information of a single-trait model is implemented in a new meta-model as follows:(5)$$y_{ij}=\beta_{1}{\hat{Z}}_{1ij}+\beta_{2}{\hat{Z}}_{2ij}+\ldots +\beta_{L}{\hat{Z}}_{Lij}+e_{ij}$$
where ${\hat{Z}}_{Lij}$ represents the scaled predictions of each trait obtained from the single-trait MGBLUP model in the first stage analysis, and $\beta_{L}$ is the β coefficients for each prediction. Each prediction was scaled by subtracting its mean $\left({\hat{\mu }}_{Lij}\right)$ and dividing by its standard deviation $\left({\hat{\sigma }}_{Lij}\right)$, that is, ${\hat{Z}}_{Lij}=\left({\hat{y}}_{Lij}-{\hat{\mu }}_{Lij}\right){\hat{\sigma }}_{Lij}^{-1}$. The BMORS model is an expansion of the multi-label classification method exploiting dependencies between target variables (e.g., multiple phenotypic traits in GP) in order to improve prediction accuracy [69,70,71]. This method captures correlations between phenotypic traits by appropriate choices of covariance functions such as the weighted regressors used in the proposed model.

#### 2.6.4. Deep Learning (DL) Models

Single-trait Multi-environment Deep Learning (SMDL) and Multi-trait Multi-environment Deep Learning (MMDL) models delineated by Montesinos-López et al. [36] were also included in prediction analyses. In brief, a densely connected neural network consisting of an input layer, multiple output layers, and multiple hidden layers between them was constructed. Then, the input variables (e.g., SNPs) were fed into the neural network and transformed by the neurons on each hidden layer with geometric non-linear functions. The final output layer is a vector of numbers (e.g., phenotypic values), or a matrix of multiple variables (e.g., multi-trait phenotypic values) predicted by the neural network. The MMDL model has multiple output neurons instead of one neuron in the SMDL model. The success of implementing DL models relies on a fine-tuning process which is involved with selecting hyperparameters including the number of neurons, number of epochs, number of layers, type of regularization, and type of action function. Based on previous studies using similar types of data [35,36,72], we included three hidden layers and used the rectified linear activation unit as an activation function and the dropout type (25% dropout rate) as the regularization method. For our study, a second-order response surface search method with a full factorial design was implemented to find the optimal combination of number of neurons and epochs for our dataset. We evaluated numbers of neurons from 5 to 70 with an increment of 5 and numbers of epochs from 10 to 80 with an increment of 10. A quadratic plateau non-linear model was used to locate the optimal number of neurons for each level of number of epochs.

### 2.7. Model Evaluation

All five predictive models were evaluated using a five-fold cross-validation (CV) approach for their prediction accuracies. Under this CV, the dataset was partitioned into five subgroups of equal size; four of the five subgroups (i.e., the training population) were used to fit each prediction model, while the remaining subgroup (i.e., the validation population) was used to assess the correlation between the observed and predicted trait values. This process was repeated five times, with each subgroup being used as the prediction set once. A stratification method was employed to evaluate the influence of population structure on prediction accuracies for all three models. Briefly, the population was split into 10 clusters based on the discriminant analysis of principal components (*DAPC*) [73] clustering approach using all 27,957 SNPs, so that a similar number of lines belonging to the same cluster were present in either the validation or training population. We also used a random partitioning method without considering the underlying population structure in the panel. For DL models, the response surface search optimization was performed before CV, and an optimal combination of number of neurons and epochs was used to compare the results with the other three models. Prediction accuracies were calculated as $r_{GY}=\;r_{p}/\sqrt{H^{2}}$, where $r_{p}$ is the mean predictive correlations across five folds. In addition, the prediction accuracy of the BMORS model was evaluated across four environments in which the dataset from each environment was predicted by the dataset from the other three environments. The model is denoted as BMORS. Finally, both the BMTME and BMOR models were implemented with 15,000 iterations, of which 10,000 were used as burn-in to fit the models.

The standard error of prediction accuracy for each environment and each model was calculated based on ${\mathrm{SE}\;}_{GYP}={\;\sigma }_{r_{p}}/\sqrt{fH^{2}}$, where $\sigma_{r_{p}}$ is the standard deviation of the predictive correlation; f is the number of folds (five in this case). Response to selection (RTS) was calculated using the formula *R* = *H^2^S* [74], where *H^2^* is the heritability for grain yield and *S* is the selection differential (in unit of kg ha^−1^). To be specific, all 237 lines were ordered according to their GEBV calculated from each model in each environment. Then, the top 10% lines were chosen as the selected population (i.e., selection intensity of 10%). The selection differential was calculated as the difference of grain yield between the means of selected lines and the whole population: *S* = $\hat{\mu }$_S_ – $\hat{\mu }$_P_, where μ_S_ is the mean yield of 10% selected lines based on GEBV and μ_P_ is the mean yield of the population. The response to selection for all three models at each environment were computed with and without correction for DTH. The mean of RTS was calculated for each environment and each model across five folds. The standard error of RTS was calculated based on ${\mathrm{SE}\;}_{GYRTS}={\;\sigma }_{RTS}/\sqrt{f}$, where $\sigma_{RTS}$ is the standard deviation of the RTS; and  f is the number of folds (five in this case).

### 2.8. Software Implementation

Phenotypic data analysis, including BLUPs and heritability calculation, and correlation analyses, were performed using R (R Development Core Team 2018). Basic models (1–2) were fit with the “lme4” package [75]. Genetic correlations between phenotypic traits were calculated using the “BMTME” package [67]. Prediction models (4) and (5) were fit with the package “BGLR” and “BMTME”, respectively [67,76]. Two DL models were evaluated using “Keras” and “tensorflow” packages [77,78]. The *DAPC* analysis was performed using an “adegenet” package [73]. The response surface search was conducted with “rsm” package [79]. Cross-validation and prediction accuracy calculation were conducted using customized codes in R.

### 2.9. Data Availability

All data generated or analyzed during this study are available in the supplemental files, including phenotypic data in “multi-trait GS phenotypic data.csv” and genotypic data in “multi-trait GS genotypic data.txt”.

## 3. Results

### 3.1. Descriptive Statistics

A description of GY, HI, SF, and TGW phenotypic traits is presented in Table 2. Phenotypic BLUEs and heritability values varied significantly among four environments. For Quincy, a generally lower temperature environment compared to Citra showed the highest BLUEs of GY, HI, and TGW (5.3 t ha^−1^, 42.7%, and 40.9 g, respectively) in 2017, compared to other three environments. For Citra, a hotter and drier environment compared to Quincy had the lowest BLUEs of GY, HI, and SF (2.0 t ha^−1^, 30.5%, and 63.9 grains/g of chaff weight, respectively) in 2018. Citra 2018 had the lowest value for TGW (34.1 g) and the highest value for SF (98.3 g). In general, Citra 2017 and Citra 2018 showed higher broad-sense heritability values than Quincy 2017 and Quincy 2018 for all four traits. For GY, Citra 2018 had the highest heritability (0.80), while Quincy 2018 had the lowest value (0.24). For HI, Citra 2017 had the highest heritability (0.78), while Quincy 2018 had the lowest value (0.26). Quincy 2017 showed the lowest heritability for SF (0.22), and Citra 2018 had the highest value (0.68). Quincy 2018 showed the lowest heritability for TGW (0.44), and Citra 2018 had the highest value (0.87). In general, a higher coefficient of variation was shown in Citra 2017 and Quincy 2017 compared to that in Citra 2018 and Quincy 2018 for all four traits.

Genetic correlations among four traits and four environments are presented in Table 3 and Table 4, respectively. The highest positive genetic correlation among traits was found between GY and HI (0.67). Relatively low genetic correlations were found between GY and SF (0.17), GY and TGW (0.18), and HI and SF (0.17). The HI and TGW had the lowest positive genetic correlation (0.10). The SF and TGW showed a negative genetic correlation (−0.32). Correlations between environments were generally low and ranged from 0.16 to 0.24. The highest and lowest correlations were found between 2017 Quincy and 2018 Quincy (0.24), and 2018 Quincy and 2018 Citra (0.16), respectively.

### 3.2. Prediction Accuracy

Population structure was determined by using the *DAPC* algorithm, and the panel was clustered into 10 groups (Figure 1). Each subgroup consisted of 14 to 38 lines, which were then randomly assigned to five different folds for cross-validation analysis. This process is considered as a stratification of both training and validation populations. 

For DL models, optimal epoch and neuron combinations were identified based on the results of response surface research (Supplement Appendix A). Then, the prediction accuracy of each trait was calculated based on the identified optimal epoch and neuron combination. When populations were not stratified or randomly sampled (noted as “un-stratified”), prediction accuracies ranged between −0.23 and 0.59 for GY, 0.07 and 0.55 for HI, 0.13 and 0.78 for SF, 0.20 and 0.88 for TGW among four environments and three models (Figure 2). Although predictive correlations of all models for Quincy 2018 were not significantly different from zero (*p* > 0.05), the low heritability of GY in this environment contributed to the negative predication accuracies in general. Overall, statistical models including MGBLUP and BMTME showed higher prediction accuracies than DL models (SMDL and MMDL). The BMOR model showed the highest prediction accuracies in the majority of the cases except for SF in Citra 2017. However, the differences between statistical models and DL models were minimal in some environments and traits. For example, DL models were comparable to statistical models for HI across environments. For SF, two DL models showed higher prediction accuracies compared to two statistical models in Quincy 2017 and Quincy 2018. When comparing results from four environments, Citra 2017 showed high prediction accuracies for all four traits. For GY, Quincy 2018 had lower prediction accuracies compared to other environments. Citra 2018 had lower prediction accuracies for SF and TGW compared to other environments. For HI, the prediction accuracies varied between models and environments. 

When populations were stratified (noted as “stratified”), prediction accuracies ranged between −0.22 and 0.62 for GY, 0.03 and 0.55 for HI, 0.16 and 0.83 for SF, and 0.21 and 0.85 for TGW among four environments and three models (Figure 3). A similar pattern was observed between the stratified and un-stratified strategy for prediction accuracy across environments (Figure 2 and Figure 3). 

For MGBLUP and BMOR models, the averaged prediction accuracies across environments were higher for TGW, which was followed by SF, HI, and GY in order (Figure 4). For the BMTME model, GY had higher prediction accuracy than HI. Prediction accuracies were not significantly affected by the stratification of populations. The SMDL and MMDL models followed the same pattern and had lower prediction accuracies than statistical models when comparing the averaged values. However, the multi-traits models including BMTME and MMDL showed higher prediction accuracies than their counter-part single-trait models for all four traits. When prediction accuracies were averaged for each model, the BMOR model showed the highest prediction accuracy followed by BMTME, MGBLUP, MMDL, and SMDL in order (Figure 5). 

We also applied the BMOR model to predict whole environments using the remaining environments as training datasets (Figure 6). Prediction accuracies ranged between 0.31 and 0.59 for GY, 0.14 and 0.54 for HI, 0.35 and 0.82 for SF, 0.54 and 0.91 for TGW.

### 3.3. Response to Selection

Response to selection (RTS) was compared in the same fashion as prediction accuracy for each model × environment combination. When populations were not stratified, RTS ranged from −0.05 to 0.5 ton ha^−1^ for GY, 0.09 to 4.94% for HI, 0.45 to 3.90 grains g^−1^ of chaff weight for SF, and 0.99 to 2.02 g for TGW among four environments and three models (Figure 7). In general, statistical models had higher RTS than DL models with exceptions of GY in Citra 2018, HI in Citra 2017, and SF in Citra 2017, Quincy 2017, and Quincy 2018. For all five models, the highest and lowest RTS for GY and HI was found in Citra 2017 and Quincy 2018, respectively. For SF, the highest and lowest RTS showed in Citra 2018 and Quincy 2017. For TGW, the highest and lowest RTS values were found in Quincy 2017 and Quincy 2018.

When populations were stratified, a similar pattern was observed for RTS compared to an un-stratified strategy. Response to selection ranged between −0.04 and 0.48 ton ha^−1^ for GY, 0.13 and 5.53% for HI, 0.26 and 4.46 for SF grains g^−1^ of chaff weight, and 1.01 g and 2.70 g for TGW among four environments and five models (Figure 8). However, the BMOR model showed significantly higher RTS for SF and TGW in Citra 2018 and Quincy 2018 compared to the other two models. 

For the average RTS of GY and HI across environments, they were not significantly different between un-stratified and stratified strategy (Figure 9). The highest average RTS for GY was found using the BMOR model with a stratified strategy (0.23 ton ha^−1^) (Figure 9). The highest average RTS for HI was found using the BMTME (1.93%) and BMOR model (1.93%) with an un-stratified strategy (Figure 9). For SF, the highest and lowest average RTS values were found using the BMOR model with a stratified strategy (3.74 grains/g of chaff weight) and the BMTME model with a stratified strategy (1.87 grains/g of chaff weight), respectively (Figure 9). For TGW, the highest and lowest average RTS values were found using the BMOR model with a stratified strategy (2.02 g) and the MMDL model with a stratified strategy (1.24 g), respectively (Figure 9). In general, the BMOR model showed the highest RTS followed by BMTME and MGBLUP in order. However, the differences of RTS among three models were smaller in magnitude when comparing to prediction accuracy. Notably, the DL models only showed higher RTS than statistical models for SF. The multi-trait models had higher RTS than single-trait models.

When applying the BMOR model to predict RTS across environments, the RTS ranged from 0.26 to 0.69 ton ha^−1^ for GY, 0.45 to 5.22% for HI, 6.52 to 13.33 grains/g of chaff weight for SF, and 3.03 to 4.36 g for TGW (Figure 10).

## 4. Discussion

In plant breeding programs, plant breeders usually perform selection for the improvement of different traits that raise the economic value of plants. When performing selection for an environment, breeders generally apply selection for several traits simultaneously associated with the most important economic traits [74]. For example, when a small grain breeder selects for GY, he also selects indirectly for other yield components, such as grain number, TGW, HI, or different physiological traits, such canopy temperature or NDVI, which are associated with grain yield. Grain number in wheat is a product of spike dry weight and grain number per unit of spike chaff weight, which is known as SF [80]. It is a major potential component of grain number m^−2^. Evidence have supported the idea of manipulating SF in breeding programs to increase sink strength and ultimately increase yield potential [46,50,53,81,82]. A strong association between SF and GY, HI, and grain number has been reported in wheat [46]. These studies suggest that the increases in SF would be related to a greater partitioning of photo-assimilates to increase GY and HI in wheat. SF has moderate heritability and is difficult and expensive to estimate as it requires spike count m^−2^, spike harvest and threshing, and separation of grain and chaff from the spike. Due to its difficulty and the cost of estimation and moderate heritability, as well as correlation with GY, HI, and grain number, SF is a perfect candidate for a multi-trait genomic selection approach to increase the predictive accuracy of GY, HI, and other associated traits [16]. Currently, plant breeding programs are mostly practicing targeted single-trait GP approaches, not considering a full exploitation of genetic information (linkage and pleiotropic effect) from correlated traits. The joint prediction of multiple traits through Multi-Trait Genomic Prediction (MTGP) approach is designed to benefit from genetic correlation between traits and the indirect selection of a target trait with relatively low heritability that genetically correlated with other high-heritability traits [19,21]. Thus, the joint multi-trait model obtained higher prediction accuracy than single-trait methods, especially for a low-heritability trait. One of the major limitations of using a multi-trait model is correlations between traits that are in practice undesirable for plant breeders. In our present study, we used four traits, and all are positively correlated with each other except for the association between SF and TGW. Thus, generally, the prediction accuracy for the different traits in our study was higher for multi-trait than single-trait genomic prediction, which is different from a previously reported study in US soft wheat [83], but it is in agreement with the results reported in European rye by Schulthess et al. [19]. Additionally, the testing environments in the present study were stressed by drought and heat, which usually makes the phenotyping of complex traits more complicated through adding environmental uncertainty. The inclusion of genotype × environment interaction in the multi-trait model improved the prediction accuracy for the joint prediction of multiple traits. The increased accuracy and RTS using a multi-trait multi-environment model for stressed environments certainly demonstrates the effectiveness of the model when the right correlated traits are included.

Our study exploited both single- and multi-trait and multi-environment models to predict yield and yield component traits including GY, HI, SF, and TGW using a diversity soft wheat panel. The results are in favor of the multi-trait and multi-environment statistical model (BMTME) for prediction accuracy and response to selection of all four traits when comparing to the single-trait and multi-environment model (MGBLUP), single-trait and multi-environment deep learning model, and multi-trait and multi-environment deep learning model. This result is in concordance to previous studies that reported that multi-trait and multi-environment GP models could be implemented to increase the prediction accuracy and RTS for low-heritability traits correlated with higher-heritability traits [16,19,83,84,85]. Jia and Jannink [16] also indicated that a multi-trait model is more effective when the genetic correlation is moderate between these traits. For prediction accuracy, traits with lower heritability such as GY showed more benefit compared to high heritability traits such as TGW using the BMTME model (46% and 11% increase, respectively). In regard to RTS, the multi-trait statistical model also showed 5 to 22% more genetic gain compared to a single-trait model across the environment from the current study. However, the benefit of the multi-trait model for RTS was varied among traits and less relevant to their heritability values based on this study compared to prediction accuracy. The deep learning models showed comparable performance to statistical models, especially for RTS. The multi-trait DL model also performed better than a single-trait DL model in most of the scenarios. Although the prediction accuracy was lower for DL models comparing to statistical models, DL models were less time consuming when computing predicted values for our dataset (23 min on average for DL models and 436 min on average for statistical models). It is also believed that a high dimensional and large dataset could benefit DL models significantly in genomic prediction [36,86]. However, it is important to recognize that the performance of DL models is highly dependent on the SNP set and phenotype. A deep learning model must be curated and calibrated specifically for traits with complex genetic structure [32].

The use of a stratified cross-validation scheme with all five models did not increase the prediction accuracy compared with using an un-stratified cross-validation scheme in the present study. One possible reason is that the alleles of representative quantitative trait loci (QTL) associated with target traits are commonly shared between training and validation populations in both stratified and un-stratified schemes. Ward et al. [83] also found that using un-related training and validation population schemes did not affect the predictive ability compared with using a related cross-validation scheme. Rutkoski et al. [84] also reported that using a multi-trait model including secondary traits had no influence on prediction accuracy if secondary trait phenotypes were not replicated in the validation test.

Although the BMOR model showed higher prediction accuracy and comparable RTS to that of the BMTME model, it does not estimate the covariances between traits and environments because it implements univariate analysis at both stages [29]. However, it is more computationally efficient than the MGBLUP and BMTME models (436 min on average) in terms of the computational resource and running time for the model to converge (42 min on average). Thus, it is advantageous when investigators are exploring the performance of genomic prediction in some preliminary studies. Therefore, we implemented the BMOR model to predict yield and yield component traits among environments. Heslot et al. [87] pointed out that GP results could be largely affected by an interaction between un-selected trait and environment being tested for selected traits, especially when selection were guided toward the un-selected traits such as stress tolerance traits with a large QTL effect. In our study, this is reflected by the inconsistent prediction accuracy and RTS when the BMOR model is applied in two cross-validation schemes and prediction among environments. For example, our soft wheat lines generally showed varying degrees of heat stress tolerance and were evaluated in Citra, FL where heat stress was common during the anthesis and grain-filling stages. The phenotype of target traits such as SF and TGW could be masked by the stress tolerance characteristics of each line. Based on our study, SF is the most affected trait, as the prediction accuracy and RTS for SF showed the largest difference between the two environments compared to other traits.

## 5. Conclusions

The study demonstrates that the multi-trait model has in general higher predictive accuracy than the single-trait model under a multiple-environmental analysis and has the capacity to predict the performance of genotypes for different test environments. It is useful for plant breeding scenarios where several economically important traits are inter-correlated. The findings of the present study could be potentially applied in plant breeding to achieve more cycles of selection by unit of time for multiple traits, to assess accurately genotype performance due to the low number of testing environments or due to a lack of replication, and to predict the performance of genotypes for stressed environments with low heritability. The analysis also showed that statistical models were superior to DL models for the studied traits, but DL models were comparable to statistical models in many cases. In conclusion, our study showed that for our population and traits of interest, multi-trait and multi-environment models can be exploited to achieve generally higher increases in prediction accuracy and RTS in several focal traits.

## Figures and Tables

**Figure 1 genes-11-01270-f001:**
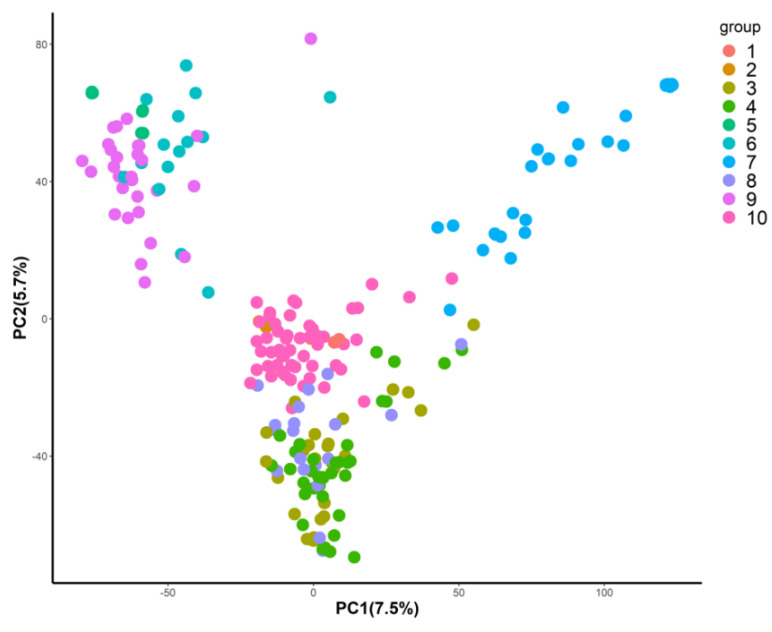
Stratification of genomic prediction panel inferred from discriminant analysis of principal components (DAPC) using 27,957 SNPs data. The first two principal components (account for variance of 7.5% and 5.7%, respectively) are used to represent each line in the genomic prediction (GP) panel. Each line was colored based on the posterior of probability assigned to 10 genetic groups inferred from DAPC.

**Figure 2 genes-11-01270-f002:**
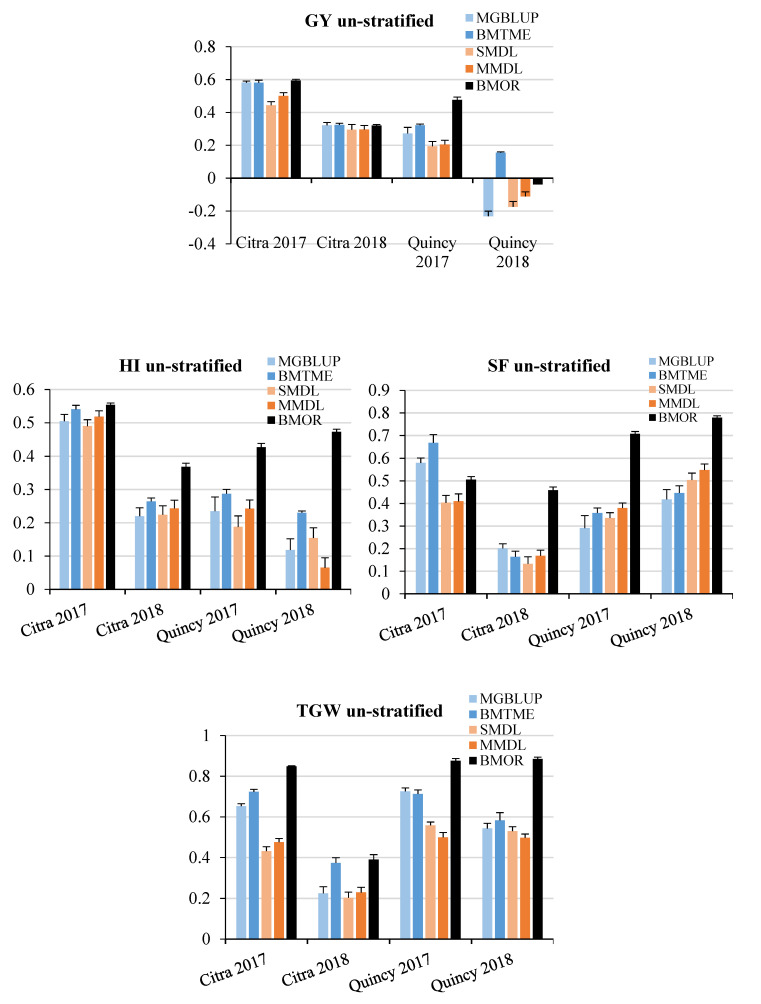
Prediction accuracies for GY, HI, SF, and TGW without population stratified. GY, grain yield; HI, harvest index; SF, spike fertility; TGW, thousand grain weight. Mean Pearson’s correlations and standard errors for each environment were presented for each trait. Statistical models were colored in light blue and blue while DL models were colored in light orange and orange. The Bayesian Multi-output Regressor (BMOR) model was colored in black.

**Figure 3 genes-11-01270-f003:**
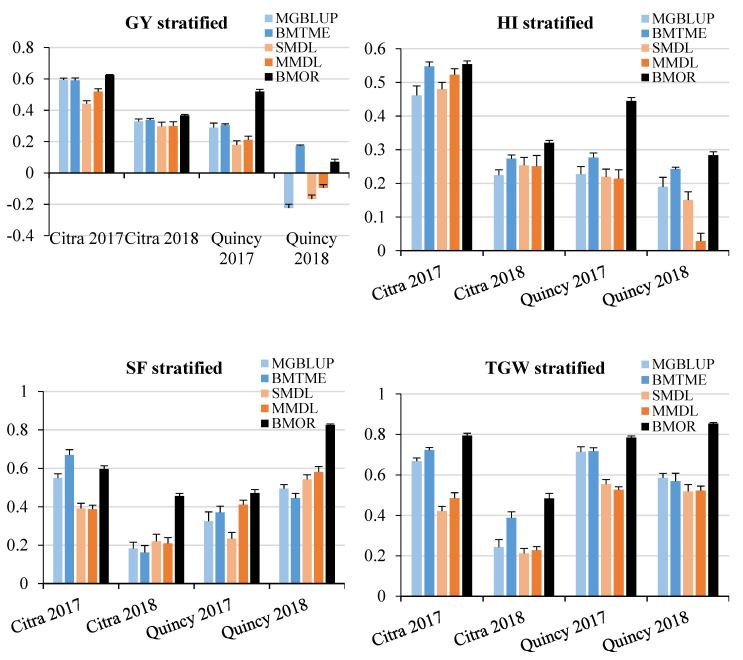
Prediction accuracies for GY, HI, SF, and TGW with population stratified. GY, grain yield; HI, harvest index; SF, spike fertility; TGW, thousand grain weight. Mean Pearson’s correlations and standard errors for each environment were presented for each trait. Statistical models were colored in light blue and blue while DL models were colored in light orange and orange. The BMOR model is colored in black.

**Figure 4 genes-11-01270-f004:**
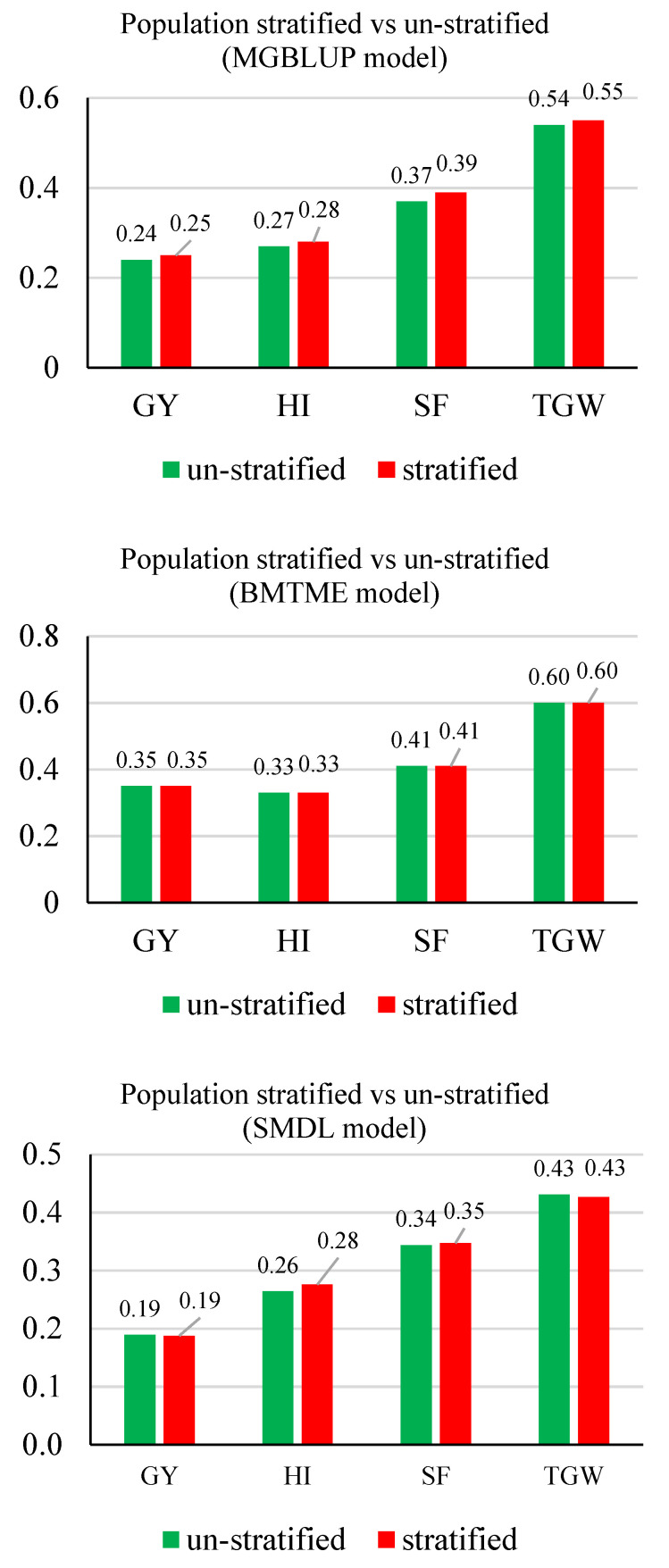
Average prediction accuracies for GY, HI, SF, and TGW with/without population stratified for five models. GY, grain yield; HI, harvest index; SF, spike fertility; TGW, thousand grain weight. Mean Pearson’s correlations for each trait were presented and labeled. The stratified scheme was colored in green, and the un-stratified scheme was colored in red.

**Figure 5 genes-11-01270-f005:**
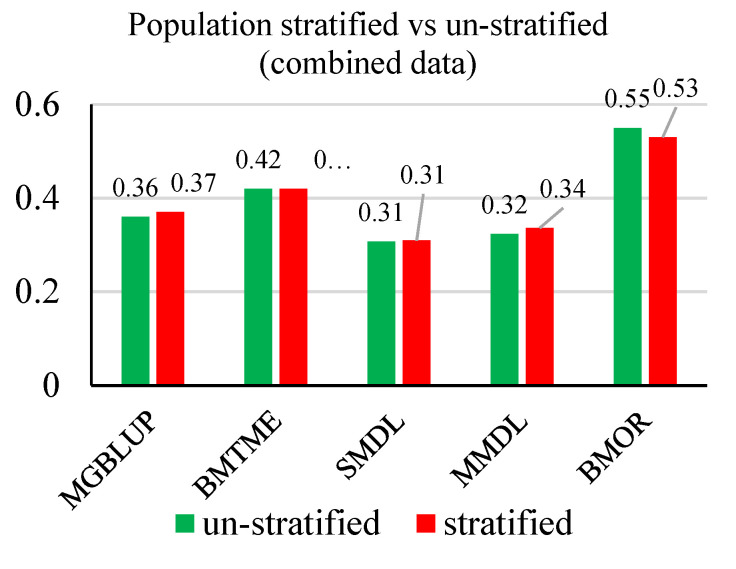
Average prediction accuracies for combined data with/without population stratified for five models Mean Pearson’s correlations for each trait were presented and labeled. The stratified scheme was colored in green, and the un-stratified scheme was colored in red.

**Figure 6 genes-11-01270-f006:**
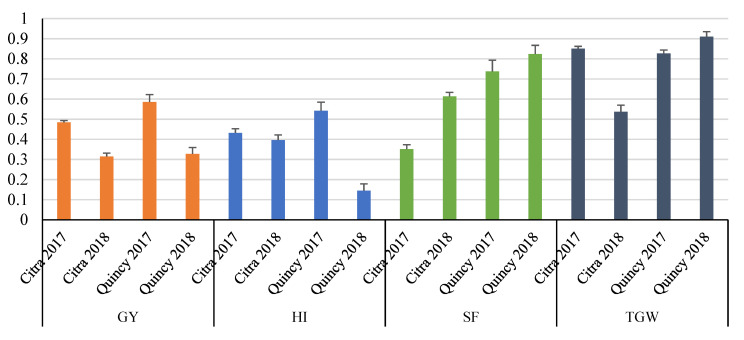
Prediction accuracies for GY (t ha^−1^), HI (%), SF (grains/g chaff weight), and TGW (g) across environments using the BMOR model. GY, grain yield; HI, harvest index; SF, spike fertility; TGW, thousand grain weight. Mean Pearson’s correlations and standard error for each environment were presented for each trait. The results of GY, HI, SF, and TGW trait were colored in orange, blue, light green, and blue-gray.

**Figure 7 genes-11-01270-f007:**
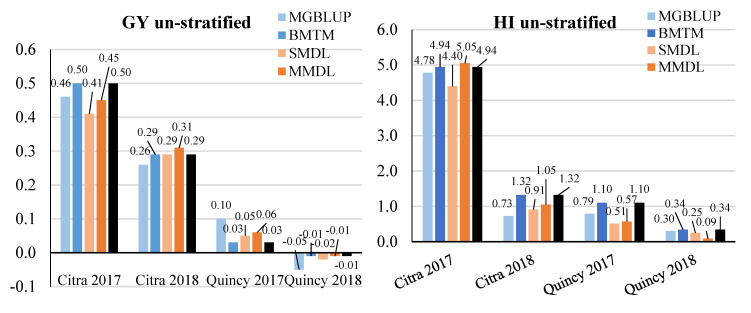
Response to selection for GY (t ha^−1^), HI (%), SF (grains/g chaff weight), and TGW (g) without population stratified. GY, grain yield; HI, harvest index; SF, spike fertility; TGW, thousand grain weight. Response to selection for each environment was presented and labeled for each trait. Statistical models were colored in light blue and blue, while DL models were colored in light orange and orange. The BMOR model was colored in black.

**Figure 8 genes-11-01270-f008:**
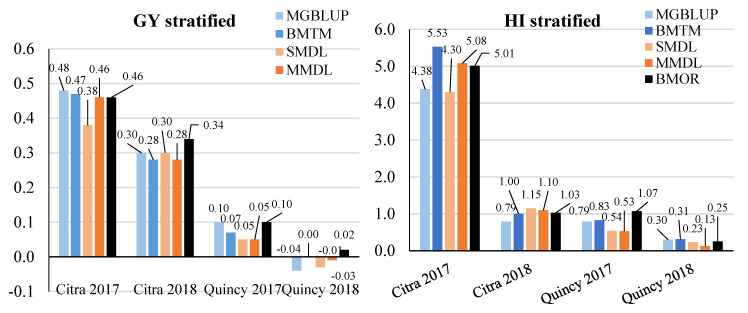
Response to selection for GY (ton ha^−1^), HI (%), SF (grains/g chaff weight), and TGW (g) with population stratified. GY, grain yield; HI, harvest index; SF, spike fertility; TGW, thousand grain weight. Response to selection for each environment was presented and labeled for each trait. Statistical models were colored in light blue and blue, while the DL models were colored in light orange and orange. The BMOR model was colored in black.

**Figure 9 genes-11-01270-f009:**
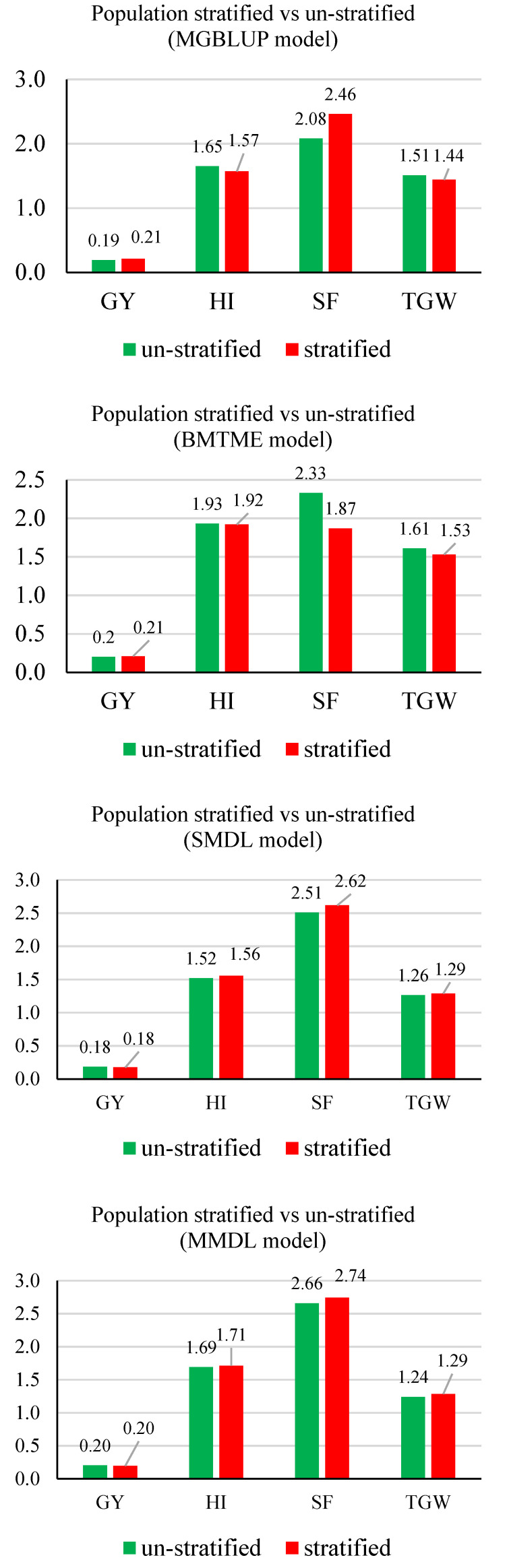
Average response to selection for GY (ton ha^−1^), HI (%), SF (grains/g chaff weight), and TGW (g) with/without population stratified for five models. GY, grain yield; HI, harvest index; SF, spike fertility; TGW, thousand grain weight. Mean response to selection for each trait were presented and labeled. Stratified scheme was colored in green and un-stratified scheme was colored in red.

**Figure 10 genes-11-01270-f010:**
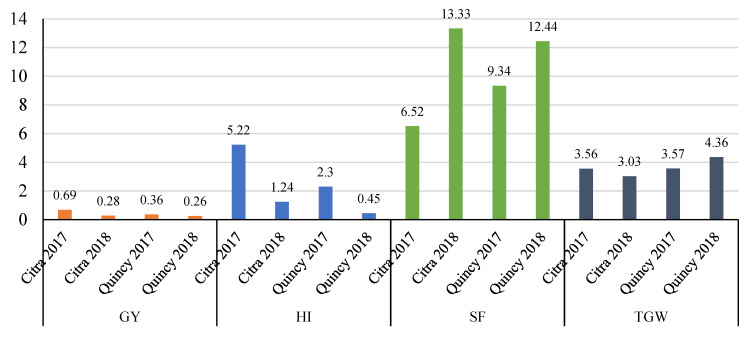
Response to selection for GY (ton ha^−1^), HI (%), SF (grains/g chaff weight), and TGW (g) across environments using the BMOR model. GY, grain yield; HI, harvest index; SF, spike fertility; TGW, thousand grain weight. Mean response to selection for each environment were presented for each trait. The results of GY, HI, SF, and TGW trait were colored in orange, blue, light green, and blue-gray.

**Table 1 genes-11-01270-t001:** Experimental site information including name of the sites, years evaluated, coordinates, and soil type.

Site	Year	Coordinates	Soil Type ^1^
Citra	2016–2017	29°24′18″ N 82°10′22″ W	Well-drained sandy soil with loamy subsoil at 20–80 inches
	2017–2018	29°24′32″ N 82°10′46′’ W	
Quincy	2016–2017	30°33′04″ N 84°35′51″ W	Well-drained loamy soils
	2017–2018	30°32′45″ N 84°35′46″ W	

^1^ Source: Soil Map of Florida (EUDASM).

**Table 2 genes-11-01270-t002:** Description of grain yield (GY) (ton ha^−1^), harvest index (HI) (%), spike fertility (SF) (grains/g of chaff weight), and thousand grain weight (TGW) (g) phenotypic traits† evaluated at Citra, FL and Quincy, FL in 2017 and 2018. The best linear unbiased estimates (BLUEs), standard error (SE), heritability (H2), coefficient of variation (CV), maximum and minimum value were calculated for each trait in four environments.

	Trait	BLUE	SE	*H* ^2^	CV	Min	Max
Citra 2017	GY	2.0	0.1	0.71	28.3	0.3	4.5
	HI	30.5	0.8	0.78	17.8	16	52
	SF	63.9	1.7	0.38	27.2	12	142
	TGW	34.7	0.4	0.48	10.8	24	48
Citra 2018	GY	3.8	0.1	0.80	11.5	1.0	7.0
	HI	37.4	0.4	0.74	6.7	20	48
	SF	98.3	1.2	0.68	9.5	62	161
	TGW	34.1	0.4	0.87	5.1	19	46
Quincy 2017	GY	3.3	0.1	0.36	16.6	1.5	5.6
	HI	34.3	0.4	0.43	12.6	20	47
	SF	83.2	1.7	0.22	25.1	34	148
	TGW	39.4	0.3	0.58	7.4	26	50
Quincy 2018	GY	5.3	0.1	0.24	18.4	2.1	8.8
	HI	42.7	0.3	0.26	9.8	28	54
	SF	94.6	1.4	0.32	15.6	52	158
	TGW	40.9	0.4	0.44	9.7	30	54

**Table 3 genes-11-01270-t003:** Estimates of averaged genetic correlation (above diagonal) and Pearson correlation of phenotypic values (below diagonal) among GY (ton ha^−1^), HI (%), SF (grains/g of chaff weight), and TGW (g) across four environments.

	GY	HI	SF	TGW
GY		0.67	0.17	0.18
HI	0.76		0.17	0.10
SF	0.36	0.30		−0.32
TGW	0.33	0.24	−0.23	

**Table 4 genes-11-01270-t004:** Estimates of genetic correlation among four environments averaged over four traits at Citra, FL and Quincy, FL in 2017 and 2018.

	Quincy 2017	Citra 2017	Quincy 2018	Citra 2018
Quincy 2017		0.24	0.26	0.19
Citra 2017			0.19	0.17
Quincy 2018				0.16

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
