# Peer review of "Multi-Trait Genomic Prediction of Yield-Related Traits in US Soft Wheat under Variable Water Regimes"

_genes, 2020, doi:10.3390/genes11111270_

Round 1
Reviewer 1 Report
In this manuscript, the authors tested multiple-traits and multiple-environment genomic prediction models for grain yield and related traits in soft winter wheat. One concern is the negative values of prediction accuracy for grain yield at Quincy in 2018. Could the authors explain why negative values of prediction accuracy were observed? In addition, I have a few minor comments and edits that may improve the manuscript.
Page 2 line 55: In the validation set of individuals, GEBV are calculated based on their genotypic information, rather than phenotypic information, right?
Page 2 line 57-59: Suggest changing to “Several empirical studies have shown that GP is effective in accelerating breeding cycles and improving genetic gains per unit of time in major crops.”
Page 2 line 64: What is the difference between genomic prediction (GP) and genomic selection (GS)? It may be better to be consistent, e.g., using GP model through the manuscript.
Figure 2, 3, 7, and 8: For some traits, “BMOR” is missed in the figure legend.
Reviewer 2 Report
The manuscript compares different multi-trait statistical models for genomic prediction of yield-related traits in soft wheat. Their report on the prediction accuracies of the different models based on 5 fold cross-validation and their genetic gain based on the response to selection.
I have only two suggestions to make, which I have detailed below:
- Page 2, line 56 - replace "phenotype" with "genotype".
- In figures 2, 3, 7 and 8, the lable for the black colour (BMOR) is missing in a few graphs.
The manuscript is well written and the methods and analysis are appropriate. The authors have applied recently developed models, including deep learning and I believe the data presented will be very useful for the wheat research/breeding community.
